# HIV Prevalence in Migrant Groups Based on Country of Origin: A Systematic Review on Data Obtained between 1993 and 2020

Cecilie Schousboe [1,*] and Christian Wejse [2]

1   Department of Public Health, Center for Global Health, Aarhus University, Bartholins Allé 2,
    8000 Aarhus, Denmark
2   Department of Infectious Disease, Aarhus University Hospital, Palle Juul-Jensens Blvd. 99,
    8200 Aarhus, Denmark; wejse@ph.au.dk
*   Correspondence: schousboe@live.dk; Tel.: +45-22268351

**Abstract:** The prevalence of internationally displaced people has been rising steadily within the last decade, creating enormous groups of migrants vulnerable to communicable diseases. This study aims to investigate HIV prevalence in migrant groups based on country of origin and present these results as weighted estimates on HIV prevalence based on geographical origin. Furthermore, HIV prevalence by country of origin is compared to WHO estimated prevalence in these countries. A systematic literature search has been conducted, and risk of bias in the included studies has been assessed. A ratio termed the Migration/Origin ratio, expressing weighted estimates on HIV prevalence among migrants by country of origin compared to the WHO estimated HIV prevalence in the country of origin, was constructed to compare the yields of this study to WHO prevalence estimates. Based on the search strategies covering the years 1990 to February 2021, 2295 articles were identified. The articles were screened by title and/or abstract, and retrieved articles were screened by full manuscript, leading to a final inclusion of 49 studies. HIV prevalence among migrants originating from the Middle East was 0.11%, Southeast Asia 1.50%, Eastern Europe 0.44%, Latin America 0.74%, North-, East-, West-, Central- and Southern Africa 1.90%, 3.69%, 2.60%, 3.75% and 3.92%, respectively. The overall Migration/Origin ratio was 2.1. HIV prevalence among migrants originating from countries with a high HIV prevalence was generally higher than among the autochthonous population. Several HIV prevalence estimates among migrants according to country of origin varied from WHO estimates.

**Keywords:** HIV prevalence; migrant health; infectious disease screening; HIV screening

## 1. Introduction

An estimated 79.5 million people worldwide were at the end of 2019 forcibly displaced from their homes due to war, persecution, or oppression. Of these, 26.0 million people were refugees, of whom 85% were hosted in developing countries. The prevalence of internationally displaced people in the world has been rising steadily since 2010, creating an enormous group of migrants worldwide vulnerable to communicable diseases [1,2].

HIV (human immunodeficiency virus) has, since its discovery in 1981, been a major recognized global public health issue, causing more than 33.0 million deaths. Worldwide, an estimated 38.0 million people were living with HIV at the end of 2019. In 2015, migrants constituted 37% of all new HIV cases in the European Union and the European Economic Area [3–6]. HIV testing policies varies significantly among recipient countries. In the US, screening is recommended to be performed on all refugees as a part of the U.S. Domestic Medical Examination for Newly Arriving Refugees, unless they decline. In the EU, national guidelines on HIV testing are diverse and tailored, though most guidelines primarily target migrants arriving from HIV high endemic countries. Research from 2014 found that only 56% of EU countries had national or sub-national guidelines for screening of newly arrived migrants for infectious diseases [7–9]. When arriving to a recipient country, HIV prevalence in the country of origin is sometimes used to allocate the individual migrant to

a specific screening program for infectious diseases. However, whether the HIV prevalence in the country of origin is representative of the prevalence among migrants is questionable. People who migrate are often younger and healthier than the general population in the country of origin. Additionally, specific risk factors such as poor living conditions, social exclusion, labour exploitation, and abuse and violence are linked to migration and may influence the HIV prevalence among migrants [10,11].

Currently, no published review has systematically investigated HIV prevalence among migrants based on country of origin. This systematic review aims to create an overview of HIV prevalence in migrant groups based on country of origin and compare it to the estimated prevalence in the country of origin based on WHO data. This is done to investigate whether HIV prevalence among migrants is equivalent to the prevalence in the country of origin, hence providing information to be considered by health authorities when allocating migrant groups to HIV screenings. Furthermore, weighted estimates on HIV prevalence among migrants from separate parts of the world will be presented. To collect as much relevant information as possible, this review covers HIV prevalence in groups defined as migrants, immigrants, refugees, and asylum seekers. In this review, the term "migrants" is used to cover all groups of displaced people.

## 2. Materials and Methods

### 2.1. Search Strategy and Study Selection

The study protocol was registered with PROSPERO (registration no.: CRD42021236018). The aim of this review was to investigate the HIV prevalence in migrant groups based on country of origin and present these results as weighted estimates on HIV prevalence in migrant groups based on geographic origin, as well as to compare the prevalence to the WHO estimated prevalence in the countries of origin. A literature search covering the time span from 1990 to February 2021 with the purpose of identifying articles with such data was conducted. After consulting a librarian, the following search strategy was applied in PubMed and Embase, respectively; In PubMed (("HIV"[Mesh]) OR ("HIV Infections"[Mesh])) AND (("Refugees"[Mesh]) OR ("Transients and Migrants"[Mesh])) (10/2-2021) and (((("Refugees"[Mesh]) OR ("Transients and Migrants"[Mesh])) OR (refugee)) OR (migrant)) AND ((("HIV"[Mesh]) OR ("HIV Infections"[Mesh])) OR (HIV)) (14/2-2021). In Embase (HIV/exp AND migrant/exp) OR refugee/exp (14-2-2021). Furthermore, relevant references and similar articles suggested in PubMed of the included articles were screened. Titles and abstracts were screened using pre-defined screening criteria, and if the required information was not available in the abstract, a further review of the full text was conducted. Original studies and reviews on HIV prevalence in migrant groups based on country of origin published between 1993 and 2020 were included (Figure 1).

The articles selected based on the screening method were met by the following inclusion criteria: To be included in this study, articles had to include data on HIV prevalence of migrants, asylum seekers, refugees or immigrants based on their country or geographical origin. In order to calculate a weighted estimate on HIV prevalence among migrants originating from a specific country, a study population with a minimum of four migrants was required. Articles featuring data on study populations consisting of specific risk groups such as sex workers, injection drug users or men who have sex with men were excluded to minimise the risk of bias. Included articles were retrieved and read in full.

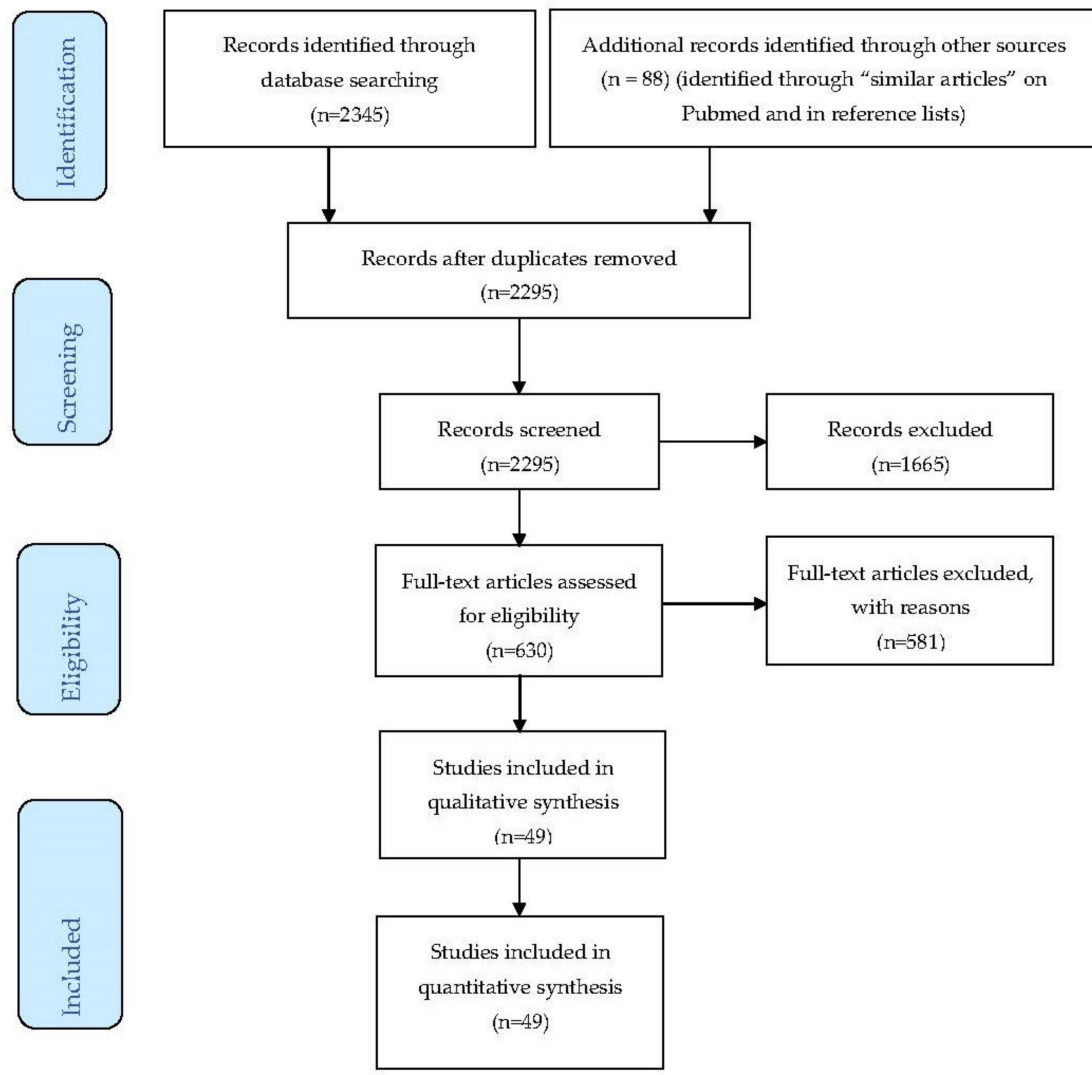

**Figure 1.** PRISMA flow diagram.

*2.2. Data Extraction and Analysis*

Data on HIV prevalence, country of origin, recipient country along with migrant status, characteristics of the study population, place and method of screening and time spent in the recipient country before being screened was extracted by simple data pooling (Table 1). Studies that reported on migrants from several countries were only included once, and data on all migrant groups was extracted. Data extraction and comparisons were accomplished according to the PRISMA statement, and the review conforms to the PRISMA checklist [12]. When more than one set of data per country was featured, a weighted estimate on HIV prevalence among migrants from this country was estimated by calculating the percentage of HIV cases based on the cumulative numerator and denominator for the relevant country. Finally, weighted estimates on HIV prevalence in migrants originating from separate parts of the world were assessed by calculating the percentage of HIV cases among the total number of screened migrants from the relevant countries. To compare the yields of the included studies to WHO prevalence estimates, a ratio termed the Migration/Origin ratio was constructed (Table 2). This ratio expressed the weighted estimates on HIV prevalence among migrants by country of origin compared to the general estimated HIV prevalence in the relevant country according to WHO data. A factor > 1 indicated a higher prevalence of HIV among migrants than what would be expected in the autochthonous population in the country of origin.

### 2.3. Risk of Bias Assessment

A risk of bias was assessed for each study using the protocol developed by Hoy et al. [13]. Each study was assessed based on ten statements, of which four evaluated the external and six the internal validity of the study in accordance with the protocol. When evaluating each study, every statement was to be responded either "yes" (0 points, low risk) or "no" (one point, high risk) to, thus indicating whether the study was at low or at high risk respectively of introducing bias. Each statement was considered after careful evaluation and a thorough review of the study in question. If information required to evaluate a question was unavailable, the risk of bias was assumed to be high. A summary item of the overall risk of bias for each study was conducted, describing the risk as either low (0–3 points), intermediate (3–6 points) or high (7–10 points) (Table 3).

### 3. Results

The literature search yielded 2295 articles that were screened by title and abstract. Of these, 1665 records were excluded because they did not feature the required information on HIV prevalence in migrant groups. This led to an assessment of eligibility of 630 articles, of which 581 were excluded since they did not meet the inclusion criteria. A total of 49 articles were included in this study, all were available in full text and published in English. Eight articles did not distinguish between specific countries but merely parts of the world of origin when accounting for HIV prevalence among migrants. Forty-four studies described serological methods of screening whereas five studies did not mention any such method. In three articles, HIV prevalence in migrants originating from Sub-Saharan Africa in general was presented. These results are not discussed further in this review, as a division of Africa into five regions was made to create a more precise overview [14]. Below, a review on HIV prevalence among migrants originating from specific countries and separate parts of the world based on the included articles is presented. Migration/Origin ratio was >1 in 34/60 of the concerned countries (Table 2). In 7/60 countries WHO estimated prevalence data was unavailable, and consequently no ratio concerning these countries could be presented. The overall Migration/Origin ratio was 2.1.

### 3.1. The Middle East

Data on HIV prevalence in migrants originating from the Middle East was extracted from 20 articles published between 2010 and 2020 [15–34]. The study populations consisted of children, pregnant women, and adults of mixed gender, respectively. The time span varied from migrants being screened for HIV upon arrival to migrants having spent more than five years in the recipient country before being screened. Eleven studies featured data on HIV prevalence among migrants originating from Afghanistan, three on Iranian migrants, seven on Iraqi migrants, one on migrants from Kurdistan, four on migrants from Pakistan, one on migrants from Palestine and six on Syrian migrants. Two studies included HIV prevalence on migrants originating from the Middle East in general. Weighted estimates on HIV prevalence according to country of origin varied from 0% to 0.4%. A total weighted estimate on the HIV prevalence among migrants originating from the Middle East based on the above-mentioned data was 0.11% with 122 cases in a study population consisting of 112,212 migrants.

### 3.2. Southeast Asia

Fourteen studies published between 2002 and 2020 included data on the HIV prevalence among migrants originating from Southeast Asia [27,29,31,32,34–43]. The study populations consisted of pregnant women, women and girls, children and adults of mixed gender, respectively. The time span varied from migrants being screened before departure to migrants having spent more than five years in the recipient country before being screened. Data on migrants originating from Sri Lanka and Vietnam, respectively, was each featured in one study. Two studies featured data on migrants originating from the Philippines and Bangladesh, and four studies included data on migrants from Myanmar.

Five studies merely included an overall prevalence of HIV among migrants originating from Southeast Asia. Weighted estimates on HIV prevalence by country of origin varied from 0% to 6.6%. A general weighted estimate on the HIV prevalence among migrants originating from Southeast Asia was 1.50% with 276 cases in a study population consisting of 18,497 migrants.

### 3.3. Eastern Europe

Nine studies published between 2008 and 2020 featured data on HIV prevalence among migrants from Eastern Europe [24,28,29,31,34,38,39,43,44]. Adults of mixed gender, children and pregnant women, respectively, constituted the study populations. The time span varied from migrants being screened before departure to migrants having spent more than five years in the recipient country before being screened. Data on migrants originating from Albania, Russia and Romania, respectively, was included in two studies, and data on migrants originating from Belarus, Georgia, Poland or Ukraine, respectively, was each featured in one study. In four studies the HIV prevalence among migrants originating from Eastern Europe in general was described. Weighted estimates on HIV prevalence according to country of origin varied from 0% to 1.4%. Forty-four cases in a study population consisting of 10,022 migrants were found, resulting in a general weighted estimate on HIV prevalence among migrants originating from Eastern Europe of 0.44%.

### 3.4. Latin America

Twelve studies published between 1999 and 2020 included data on HIV prevalence in migrants originating from Latin America [32,39,42,44–52]. The study populations were composed of adults of mixed gender, children, women and men, respectively. The time span varied from migrants being screened upon arrival to migrants having spent 17 years in the recipient country before being screened. Data on migrants originating from Bolivia, Cuba, El Salvador, Guatemala, Haiti, Honduras, Nicaragua or Suriname, respectively, was featured in one study, and three studies included data on migrants from Mexico. One study included data on migrants from the Antilles, one included migrants from Caribbean and three articles merely included data on migrants originating from Latin America in general. Weighted estimates on HIV prevalence based on country of origin varied from 0% to 2.4%. A general weighted estimate on the HIV prevalence among migrants originating from Latin America was 0.74%, with 141 cases in a study population of 19,006 migrants.

### 3.5. North Africa

Four studies published between 2012 and 2018 included data on the HIV prevalence among migrants originating from North Africa [32,38,43,53]. The study populations were comprised of adults of mixed gender and pregnant women, respectively. The time span between arriving in the recipient country and being screened for HIV varied from less than one to more than five years. Data on migrants originating from Tunisia or Morocco was mentioned in one study, respectively, and three studies included data on migrants originating from North Africa in general. The HIV prevalence among migrants from Morocco and Tunisia was 0%. Twelve cases in a study population of 633 migrants led to a general weighted estimate on the HIV prevalence among migrants originating from North Africa of 1.90%.

### 3.6. East Africa

Sixteen studies published between 1994 and 2020 included data on the HIV prevalence among migrants originating from East Africa [24,25,27–29,32,33,39,54–61]. The study population consisted of adults of mixed gender, children, women, and men, respectively. Time spent in the recipient country before being screened for HIV varied from migrants being screened before or upon arrival to a time span of five years. Eight studies included data on migrants from Eritrea, four included migrants from Ethiopia or Uganda respectively, four included migrants from Rwanda or Somalia, respectively. Three studies included migrants

from Sudan and one included migrants from Tanzania and South Sudan, respectively. Two studies featured data on migrants originating from East Africa in general. Weighted estimates on HIV prevalence by country of origin varied from 0% to 4.1%. A general weighted estimate on the HIV prevalence among migrants originating from East Africa was 3.69%, with 8311 cases in a study population consisting of 225,234 migrants.

### 3.7. West Africa

Ten studies published between 1993 and 2020 featured data on the HIV prevalence among migrants from West Africa [24–27,32,34,39,47,59,62]. The study populations consisted of adults of mixed gender and women, respectively. The time span varied from migrants being screened upon arrival to migrants having spent more than five years in the recipient country before being screened. Data on migrants originating from Gambia, Senegal and Sierra Leone was featured in four studies, on migrants from Guinea, Liberia or Togo respectively in two studies, on migrants from Ghana in three studies and on migrants from Côte d'Ivoire and Mali or, respectively, in five studies. one study included data on migrants from Guinea-Bissau and seven included data on migrants from Nigeria. Two studies merely described the HIV prevalence among migrants originating from West Africa in general. Weighted estimates on HIV prevalence by country of origin varied from 0.9% to 25%. A general weighted estimate on the HIV prevalence among migrants originating from West Africa was 2.6%, with 322 cases out of 12,275 migrants.

### 3.8. Central Africa

Eight studies published between 2006 and 2020 included data on migrants from Central Africa [24,27,33,44,59–61,63]. The study populations consisted of adults of mixed gender, children, women, and men respectively. Time spent in the recipient country before being screened varied from less than one year to five years. Three studies featured data on HIV prevalence among migrants from Burundi, two on migrants from Cameroon or Congo, respectively, one on migrants from Chad or Equatorial Guinea and five on migrants from the Democratic Republic of Congo (DRC). Weighted estimates on HIV prevalence varied from 2.6% to 8.2%. 7661 cases in a study population of 204,308 migrants led to a general weighted estimate of HIV prevalence among migrants originating from Central Africa of 3.75%.

### 3.9. Southern Africa

Four studies published between 2006 and 2017 included data on the HIV prevalence among migrants from Southern Africa [59–61,64]. Study populations consisted of adults of mixed gender, women, and men respectively. The time span varied from migrants having spent less than six months to more than five years in the recipient countries before being screened. Data on migrants originating from Angola, Malawi, Mozambique, or Zambia, respectively, was featured in one study each, and data on migrants from Zimbabwe was included in two studies. Weighted estimates on HIV prevalence based on country of origin varied from 1.8% to 24.3%. A general weighted estimate on the HIV prevalence among migrants originating from Southern Africa was 3.92%, with 7878 cases in a study population consisting of 201,014 migrants.

**Table 1.** Study characteristics.

| Study | Country of Origin | HIV Prevalence | Number of Cases/Populations Screened | Study Population Characteristics | Country and Location of Screening | Median Time Spent in Recipient Country before Being Screened |
|---|---|---|---|---|---|---|
| Ackermann et al. (2018) [24] | Eritrea | 1.2 % | 49/4068 | Asylum seekers. Mixed gender, adults. | Germany. Mandatory screening of all first-time asylum seekers. | <1 year |
| | Ethiopia | 1.4% | 26/1869 | | | |
| | Somalia | 0.5% | 1/2396 | | | |
| | Uganda | 4.3% | 6/139 | | | |
| | Congo | 6.4% | 7/110 | | | |
| | DRC | 11.1% | 7/63 | | | |
| | Mali | 0.9% | 6/695 | | | |
| | Nigeria | 2.3% | 83/3629 | | | |
| | Senegal | 1.0% | 18/1789 | | | |
| | Sierra Leone | 3.0% | 13/440 | | | |
| | Afghanistan | 0.05% | 8/16,227 | | | |
| | Iran | 0.4% | 8/1940 | | | |
| | Pakistan | 0.2% | 9/4502 | | | |
| | Georgia | 0.6% | 2/360 | | | |
| | Iraq | 0.02% | 2/8185 | | | |
| | Syria | 0.03% | 9/30,450 | | | |
| | Russia | 1.4% | 4/293 | | | |
| | Ukraine | 1.2% | 28/2303 | | | |
| | Albania | 0.1% | 5/4479 | | | |
| Alberer et al. (2018) [25] | Afghanistan | 0% | 0/108 | Refugees & Asylum seekers. Mixed gender, adults. | Germany. Reception center "Bayernkaserne" | N/A |
| | Eritrea | 0% | 0/79 | | | |
| | Nigeria | 1.8% | 2/109 | | | |
| | Sierra Leone | 0% | 0/52 | | | |
| | Somalia | 0% | 0/144 | | | |
| | Syria | 0% | 0/60 | | | |
| Angeletti et al. (2016) [23] | Syria | 0% | 0/30 | Migrant/refugee. Mixed gender, adults. | Italy. Asylum seekers center. | Upon arrival in recipient country |
| Ansari et al. (2011) [16] | Afghanistan | 5.2% | 29/556 | Refugees. Mixed gender, adults. | Pakistan. AC & free health camps. | N/A |
| Bahat et al. (2019) [21] | Syria | 0.03% | 4/11,015 | Migrants. Pregnant women. | Turkey. Hospital. | N/A |
| Baltazar et al. (2015) [64] | Mozambique | 22.3% | 71/318 | Immigrants. Men. | South Africa. N/A | >12 months |
| Barnett et al. (2013) [39] | Caribbean | 0.8% | 4/522 | Migrants. N/A | USA. Geosentinel Clinics. | <120 days |
| | East Africa | 1% | 18/1868 | | | |
| | Eastern Europe | 0% | 0/1474 | | | |
| | Southeast Asia | 0.3% | 6/2229 | | | |
| | West Africa | 0.5% | 5/658 | | | |
| Buonfrate et al. (2018) [62] | Côte d'Ivoire | 8.3% | 2/24 | Asylum seekers. Mixed gender, adults. | Italy. Refugee shelters. | <6 months |
| | Gambia | 5.0% | 2/40 | | | |
| | Mali | 0.97% | 1/103 | | | |
| | Guinea-Bissau | 25% | 1/4 | | | |
| Centers for Disease Control and Prevention (CDC) (2010) [20] | Iraq | 0.4% | 1/274 | Asylees. Mixed gender, adults. | USA. N/A | <90 days |
| Chernet et al. (2018) [56] | Eritrea | 0% | 0/107 | Refugees. Mixed gender, adults. | Switzerland. Refugee Centers. | <24 months |
| Ciccozzi et al. (2018) [54] | Eritrea | 0.7% | 1/133 | Migrant/refugee. Mixed gender, adults. | Italy. University Campus. | Upon arrival in recipient country |

**Table 1.** *Cont.*

| Study | Country of Origin | HIV Prevalence | Number of Cases/Populations Screened | Study Population Characteristics | Country and Location of Screening | Median Time Spent in Recipient Country before Being Screened |
|---|---|---|---|---|---|---|
| Coppola et al. (2020) [34] | Nigeria | 2% | 13/637 | Immigrants. Mixed gender, adults. | Italy. Clinical centers. | <7 months |
| | Ghana | 2.9% | 9/307 | | | |
| | Gambia | 0.7% | 2/300 | | | |
| | Senegal | 1.4% | 4/285 | | | |
| | Mali | 0.4% | 1/229 | | | |
| | Ivory Coast | 3.4% | 7/203 | | | |
| | Bangladesh | 0.4% | 1/282 | | | |
| | Pakistan | 0.4% | 1/251 | | | |
| | Romania | 1.1% | 2/178 | | | |
| Crawshaw et al. (2018) [33] | Afghanistan | 0% | 0/56 | Refugees Mixed gender, adults. | UK. Pre-entry health assessment. | Pre-arrival at recipient country |
| | DRC | 3.6% | 18/504 | | | |
| | Eritrea | 0% | 0/52 | | | |
| | Ethiopia | 1.5% | 4/259 | | | |
| | Iran | 0% | 0/14 | | | |
| | Iraq | 0.2% | 1/462 | | | |
| | Palestine | 0% | 0/25 | | | |
| | Somalia | 0.6% | 3/499 | | | |
| | South Sudan | 0% | 0/35 | | | |
| | Sudan | 1.5% | 5/329 | | | |
| | Syria | 0% | 0/6245 | | | |
| | Uganda | 0% | 0/1 | | | |
| Cuomo et al. (2019) [26] | Nigeria | 1.3% | 1/78 | Migrants. Mixed gender, adults. | Italy. IDC. | Upon arrival in recipient country |
| | Gambia | 0% | 0/71 | | | |
| | Mali | 0% | 0/41 | | | |
| | Pakistan | 0% | 0/41 | | | |
| | Côte d'Ivoire | 13.3% | 4/30 | | | |
| | Senegal | 0% | 0/27 | | | |
| Ditton & Lehane (2009) [40] | Myanmar | 10.7% | 120/1122 | Migrants. Mixed gender, adults. | Thailand. Hospital. | N/A |
| Doherty et al. (2020) [37] | Myanmar | 1.9% | 3/156 | Refugees. Mixed gender, adults. | Bangladesh. Refugee camps. | >5 years |
| Donate et al. (2005) [46] | Mexico | 0% | 0/1041 | Migrants. Mixed gender, adults. | Mexico, by the USA boarder. N/A | At border crossing to the USA |
| Donisi et al. (2020) [27] | Afghanistan | 0% | 0/9 | Asylum seekers. Mixed gender, adults. | Italy. Migration Health unit. | N/A |
| | Bangladesh | 0% | 0/22 | | | |
| | Côte d'Ivoire | 0% | 0/20 | | | |
| | Eritrea | 0% | 0/1 | | | |
| | Gambia | 0% | 0/35 | | | |
| | Ghana | 0% | 0/9 | | | |
| | Guinea | 0% | 0/13 | | | |
| | Liberia | 0% | 0/2 | | | |
| | Mali | 0% | 0/15 | | | |
| | Nigeria | 1.9% | 2/103 | | | |
| | Pakistan | 0% | 0/64 | | | |
| | DRC | 0% | 0/1 | | | |
| | Senegal | 0% | 0/15 | | | |
| | Sierra Leone | 0% | 0/2 | | | |
| | Togo | 0% | 0/4 | | | |
| Fuster et al. (2020) [45] | Haiti | 2.4% | 12/498 | Immigrants. Mixed gender, adults. | Chile. N/A | N/A |
| Giorgio et al. (2017) [60] | Zimbabwe | 15.5% | 27/174 | Migrants. Men. | South Africa. N/A | 1–>5 years |
| | Congo/DRC | 1.2% | 3/250 | | | |
| | Malawi | 24.3% | 6/25 | | | |
| | Tanzania | 3.8% | 7/184 | | | |

**Table 1.** *Cont.*

| Study | Country of Origin | HIV Prevalence | Number of Cases/Populations Screened | Study Population Characteristics | Country and Location of Screening | Median Time Spent in Recipient Country before Being Screened |
|---|---|---|---|---|---|---|
| Goosen et al. (2015) [59] | Angola | 1.8% | 9/497 | Asylum seekers. Women. | The Netherlands. Electronic medical records database from MOA. | <6 weeks |
| | Burundi | 8.3% | 9/108 | | | |
| | Cameroon | 13.2% | 5/38 | | | |
| | DRC | 3.1% | 8/254 | | | |
| | Eritrea | 2.3% | 1/44 | | | |
| | Guinea-Conakry | 3.9% | 6/154 | | | |
| | Côte d'Ivoire | 7.8% | 4/51 | | | |
| | Liberia | 5.8% | 4/69 | | | |
| | Nigeria | 1.1% | 1/87 | | | |
| | Rwanda | 17% | 8/47 | | | |
| | Sierra Leone | 3.9% | 10/257 | | | |
| | Somalia | 1.7% | 7/401 | | | |
| | Sudan | 1.7% | 3/180 | | | |
| | Togo | 4.1% | 2/49 | | | |
| Gras et al. (1999) [47] | Suriname | 0.4% | 3/734 | Immigrants. Mixed gender, adults. | The Netherlands. Approach at public places. | 17 years |
| | Antilles | 1.8% | 4/225 | | | 9 years |
| | Ghana | 1.4% | 7/518 | | | 6 years |
| | Nigeria | 1.4% | 1/73 | | | 2 years |
| Hall et al. (2020) [36] | Philippines | 0% | 0/1164 | Migrants. Women. | China. NGO study field site. | 39 months |
| Jabbari et al. (2011) [15] | Afghanistan | 0.2% | 1/477 | Immigrants. Mixed gender, adults. | Iran. Homes of immigrants. | N/A |
| Jackson et al. (2016) [51] | Bolivia | 0.6% | 3/486 | Immigrants Mixed gender, adults & children | Italy. Primary health center. | 4.5 years |
| Khanani et al. (2010) [18] | Afghanistan | 5.93% | 33/556 | Refugees. Mixed gender, adults. | Pakistan. AC and free health camps. | N/A |
| Kissinger et al. (2012) [49] | Honduras | 0% | 0/89 | Migrants. Men. | USA. N/A | 1.23 years |
| | Mexico | 0% | 0/14 | | | |
| | Guatemala | 0% | 0/8 | | | |
| | El Salvador | 0% | 0/7 | | | |
| | Nicaragua | 0% | 0/7 | | | |
| Köse et al. (2017) [22] | Syria | 2.2% | 2/88 | Refugees. Children. | Turkey. Hospital. | N/A |
| Kumar et al. (2020) [30] | Iraq | 0.1% | 1/1027 | Immigrants Mixed gender, adults & children respectively. | USA. Medical screening examination. | <90 days |
| | Iraq | 0.2% | 1/662 | | | |
| | Afghanistan | 0.06% | 3/4755 | | | |
| | Afghanistan | 0% | 0/3262 | | | |
| Laganá et al. (2015) [38] | Sri Lanka | 0% | 0/140 | Migrants. Pregnant women. | Italy. Outpatient clinic. | N/A |
| | Philippines | 0% | 0/52 | | | |
| | Morocco | 0% | 0/45 | | | |
| | Romania | 2.5% | 1/40 | | | |
| | Poland | 0% | 0/7 | | | |
| | Tunisia | 0% | 0/4 | | | |
| Manzardo et al. (2008) [44] | Cameroon | 6.3% | 6/96 | Immigrants Mixed gender, adults & children. | Spain. TMU. | <5 years |
| | Equatorial Guinea | 6.0% | 19/317 | | | |

**Table 1.** *Cont.*

| Study | Country of Origin | HIV Prevalence | Number of Cases/Populations Screened | Study Population Characteristics | Country and Location of Screening | Median Time Spent in Recipient Country before Being Screened |
|---|---|---|---|---|---|---|
| Martinez-Donate et al. (2015) [50] | Latin America | 2.1% | 7/340 | Migrants. Mixed gender, adults. | Mexico. Public sites. | N/A |
| | Eastern Europe | 1.4% | 1/72 | | | |
| | Mexico | 0.6% | 18/2811 | | | |
| McCarthy et al. (2013) [32] | East Africa | 15% | 184/1253 | Migrants. Mixed gender, adults. | International study. GeoSentinel Clinic. | <1 year–>5 years |
| Monge-Maillo (2015) [48] | Southeast Asia | 4% | 53/1200 | Immigrants. Mixed gender, adults. | Spain. TMU. | 14 months |
| | West Africa | 10% | 102/1048 | | | |
| | Middle East | 1% | 9/844 | | | |
| | South America | 4% | 31/698 | | | |
| | North Africa | 2% | 12/503 | | | |
| | Sub-Saharan Africa | 2.3% | 7/317 | | | |
| | Latin America | 0.3% | 1/383 | | | |
| Naemabadi et al. (2019) [17] | Afghanistan | 0% | 0/339 | Immigrants. Children. | Iran. 2 supporting centers. | Born in Iran |
| O'Laughlin et al. (2015) [63] | Uganda | 8.9 % | 218/2457 | Regufees. Mixed gender, adults. | Uganda. Nakivale Refugee Settlement. | 4 years |
| | Rwanda | 2.3% | 56/2395 | | | |
| | DRC | 1.9% | 30/1580 | | | |
| | Burundi | 1.4% | 14/987 | | | |
| Plewes et al. (2008) [35] | Myanmar | 0.4 % | 2/500 | Refugees. Pregnant women. | Thailand. Refugee camp. | N/A |
| Pollack et al. (1994) [57] | Ethiopia | 2.1% | 124/5800 | Immigrants. Mixed gender, adults & children. | Israel. N/A | 1–3 years |
| Rey et al. (1995) [58] | Rwanda | 4.9% | 7/143 | Refugees. Children. | DRC. Test of orphans by orphanage staff. | N/A |
| Salas-Coronas et al. (2018) [53] | Sub-Saharan Africa | 1.4% | 7/488 | Migrants. Mixed gender, adults. | Spain. TMU. | <12 months |
| | North Africa | 0% | 0/35 | | | |
| Seagle et al. (2020) [52] | Cuba | 0.5% | 58/10,753 | Migrants/refugees. Mixed gender, adults. | USA. Texas Department of State Health Services database. | N/A |
| Stauffer et al. (2012) [31] | Middle East | 0% | 0/226 | Refugees. Mixed gender, adults. | USA. Medical screening examination. | Pre-arrival at recipient country |
| Sudhinaraset et al. (2019) [42] | Eastern Europe | 0% | 0/397 | Refugees. Women and girls. | USA. RHEIS. | <90 days of arrival |
| | Southeast Asia | 0.3% | 7/2173 | | | |
| | Sub-Saharan Africa | 3.3% | 129/3964 | | | |
| | Latin America | 0.8% | 3/390 | | | |
| | Southeast Asia | 0.05% | 4/9118 | | | |
| Tiittala et al. (2018) [28] | Kurdistan | 0% | 0/415 | Migrants. Mixed gender, adults. | Finland. MAAMU. | >1 year |
| | Russia | 0% | 0/324 | | | |
| | Somalia | 0% | 0/261 | | | |
| Tramuto et al. (2012) [43] | Eastern Europe | 1.6% | 1/62 | Migrants. Mixed gender, adults | Italy. Migration medicine ambulatory. | N/A |
| Williams et al. (2020) [29] | North Africa | 0% | 0/46 | Asylum seekers. Children. | UK. 2 pediatric IDC. | 6 months |
| | Southeast Asia | 1.5% | 1/68 | | | |
| | Afghanistan | 0% | 0/43 | | | |
| | Albania | 0% | 0/32 | | | |

**Table 1.** *Cont.*

| Study | Country of Origin | HIV Prevalence | Number of Cases/Populations Screened | Study Population Characteristics | Country and Location of Screening | Median Time Spent in Recipient Country before Being Screened |
|---|---|---|---|---|---|---|
| | Eritrea | 0% | 0/43 | | | |
| | Ethiopia | 0% | 0/20 | | | |
| | Iran | 0% | 0/5 | | | |
| | Iraq | 0% | 0/9 | | | |
| | Sudan | 0% | 0/27 | | | |
| | Vietnam | 0% | 0/21 | | | |
| Wiwanitkit & Waenlor (2002) [41] | Myanmar | 3.2% | 8/250 | Migrants. Mixed gender, adults. | Thailand. District Hospital. | <1 month |
| Yanni et al. (2012) [19] | Iraq | 0% | 0/18,990 | Refugees. Mixed gender, adults & children. | Jordan. IOM clinics. | Before arrival at recipient country |
| Zencovich et al. (2006) [61] | Zimbabwe | 4.3% | 4295/100,000 | Immigrants. Mixed gender, adults. | Canada. Immigration mandatory HIV testing. | N/A |
| | Burundi | 4.3% | 4281/100,000 | | | |
| | Rwanda | 4.1% | 4103/100,000 | | | |
| | Uganda | 3.7% | 3698/100,000 | | | |
| | Zambia | 3.5% | 3470/100,000 | | | |
| | Chad | 3.3% | 3265/100,000 | | | |

**Table 2.** Weighted estimates of HIV prevalence by country of origin and WHO estimated HIV prevalence.

| Country | Total No. of Cases/Number of Migrants Screened | Weighted Estimate of HIV Prevalence | Prevalence of HIV among Adults 15–49 Years of Age According to WHO (2018–2019) [65] | Migration/ Origin Ratio |
|---|---|---|---|---|
| Afghanistan | 74/26,388 | 0.3% | <0.1% | 3 |
| Albania | 5/4511 | 0.1% | N/A | N/A |
| Angola | 9/497 | 1.8% | 2.0% | 0.9 |
| Bangladesh | 1/304 | 0.3% | <0.1% | 3 |
| Bolivia | 3486 | 0.6% | 0.3% | 2 |
| Burundi | 4304/101,095 | 4.3% | 1.0% | 4.3 |
| Cameroon | 11/134 | 8.2% | 3.6% | 2.3 |
| Chad | 3265/100,000 | 3.3% | 1.3% | 2.5 |
| Congo | 10/360 | 2.8% | 2.6% | 1.1 |
| Côte d'Ivoire | 17/328 | 5.2% | 2.6% | 2 |
| Cuba | 58/10,753 | 0.5% | 0.40% | 1.3 |
| Democratic Republic of Congo (DRC) | 63/2402 | 2.6% | 0.8% | 3.3 |
| El Salvador | 0/7 | 0.0% | 0.6% | 0 |
| Equatorial Guinea | 19/317 | 6.0% | 7.1% | 0.8 |
| Eritrea | 51/4527 | 1.1% | 0.7% | 1.6 |
| Ethiopia | 154/7948 | 1.9% | 1.0% | 1.9 |
| Gambia | 4/446 | 0.9% | 1.9% | 0.5 |
| Georgia | 2/360 | 0.6% | 0.4% | 1.5 |
| Ghana | 16/834 | 1.9% | 1.7% | 1.1 |
| Guatemala | 0/8 | 0.0% | 0.4% | 0 |
| Guinea | 6/167 | 3.6% | 1.4% | 2.6 |
| Guinea-Bissau | 1/4 | 25.0% | 3.5% | 7.1 |
| Haiti | 12/498 | 2.4% | 2.0% | 1.2 |
| Honduras | 0/89 | 0.0% | 0.3% | 0 |
| Iran | 8/1959 | 0.4% | 0.1% | 4 |
| Iraq | 6/29,609 | 0.02% | N/A | N/A |
| Kurdistan | 0/415 | 0.0% | N/A | N/A |
| Liberia | 4/71 | 5.6% | 1.3% | 4.3 |
| Malawi | 6/25 | 24.3% | 9.2% | 2.6 |
| Mali | 8/1083 | 0.7% | 1.4% | 0.5 |
| Mexico | 18/3866 | 0.5% | 0.2% | 2.5 |
| Morocco | 0/45 | 0.0% | <0.1% | 0 |
| Mozambique | 71/318 | 22.3% | 12.6% | 1.8 |
| Myanmar | 133/2028 | 6.6% | 0.8% | 8.3 |

**Table 2.** *Cont.*

| Country | Total No. of Cases/Number of Migrants Screened | Weighted Estimate of HIV Prevalence | Prevalence of HIV among Adults 15–49 Years of Age According to WHO (2018–2019) [65] | Migration/ Origin Ratio |
|---|---|---|---|---|
| Nicaragua | 0/7 | 0.0% | 0.2% | 0 |
| Nigeria | 103/4716 | 2.2% | 1.5% | 1.5 |
| Pakistan | 10/4858 | 0.2% | 0.1% | 2 |
| Palestine | 0/25 | 0.0% | N/A | N/A |
| Philippines | 0/1216 | 0.0% | 0.1% | 0 |
| Poland | 0/7 | 0.0% | N/A | N/A |
| Romania | 3/218 | 1.4% | 0.1% | 14 |
| Russia | 4/617 | 0.7% | N/A | N/A |
| Rwanda | 4174/102,585 | 4.1% | 2.5% | 1.6 |
| Senegal | 22/2116 | 1.0% | 0.4% | 2.5 |
| Sierra Leone | 23/751 | 3.1% | 1.5% | 2.1 |
| Somalia | 11/3701 | 0.3% | 0.1% | 3 |
| South Sudan | 0/35 | 0.0% | 2.5% | 0 |
| Sri Lanka | 0/140 | 0.0% | <0.1% | 0 |
| Sudan | 8/536 | 1.5% | 0.2% | 7.5 |
| Suriname | 3/734 | 0.4% | 1.4% | 0.3 |
| Syria | 15/47,887 | 0.03% | <0.1% | 3 |
| Tanzania | 7/184 | 3.8% | 4.6% | 0.8 |
| Togo | 2/53 | 3.8% | 2.3% | 1.7 |
| Tunisia | 0/4 | 0.0% | <0.1% | 0 |
| Uganda | 3704/102,597 | 3.6% | N/A | N/A |
| Ukraine | 28/2303 | 1.2% | 1.0% | 1.2 |
| Vietnam | 0/21 | 0.0% | 0.3% | 0 |
| Zambia | 3470/100,000 | 3.5% | 11.3% | 0.3 |
| Zimbabwe | 4322/100,174 | 4.3% | 12.7% | 0.3 |

N/A = Not available.

### 3.10. Quality Assessment of Included Studies

A summary of the risk of bias assessment of the included studies is presented in Table 3. Risk of bias was assessed to be low in 38 studies (78%), intermediate in 10 studies (20%) and high in one study (2%). Time spent in the recipient countries before being screened for HIV and the number of migrants screened, particularly contributed to the risk of bias. Furthermore, the representativity of the study populations in relation to the national populations as well as representativity of the sampling frames compared to the target populations caused an increased risk of bias.

**Table 3.** Risk of bias assessment [13].

| Study | External Validity | | | | | Internal Validity | | | | | SUM |
|---|---|---|---|---|---|---|---|---|---|---|---|
| | 1 | 2 | 3 | 4 | 5 | 6 | 7 | 8 | 9 | 10 | |
| Jabbari et al. [15] | | | | | | | | | | | |
| Ansari et al. [16] | | | | | | | | | | | |
| Naemabadi et al. [17] | | | | | | | | | | | |
| Khanani et al. [18] | | | | | | | | | | | |
| Plewes et al. [35] | | | | | | | | | | | |
| Hall et al. [36] | | | | | | | | | | | |
| Fuster et al. [45] | | | | | | | | | | | |
| Yanni et al. [19] | | | | | | | | | | | |
| Centers for Disease Control and Prevention (CDC) [20] | | | | | | | | | | | |
| Donate et al. [46] | | | | | | | | | | | |
| Bahat et al. [21] | | | | | | | | | | | |
| Doherty et al. [37] | | | | | | | | | | | |
| Köse et al. [22] | | | | | | | | | | | |
| Angeletti et al. [23] | | | | | | | | | | | |
| Ciccozzi et al. [54] | | | | | | | | | | | |
| O'Laughlin et al. [63] | | | | | | | | | | | |
| Chernet et al. [56] | | | | | | | | | | | |
| Pollack et al. [57] | | | | | | | | | | | |
| Gras et al. [47] | | | | | | | | | | | |

**Table 3.** *Cont.*

| Study | External Validity | | | | Internal Validity | | | | | | SUM |
|---|---|---|---|---|---|---|---|---|---|---|---|
| | 1 | 2 | 3 | 4 | 5 | 6 | 7 | 8 | 9 | 10 | |
| Baltazar et al. [64] | | | | | | | | | | | |
| Rey et al. [58] | | | | | | | | | | | |
| Ackermann et al. [24] | | | | | | | | | | | |
| Goosen et al. [59] | | | | | | | | | | | |
| Alberer et al. [25] | | | | | | | | | | | |
| Cuomo et al. [26] | | | | | | | | | | | |
| Buonfrate et al. [62] | | | | | | | | | | | |
| Giorgio et al. [60] | | | | | | | | | | | |
| Zencovich et al. [61] | | | | | | | | | | | |
| Donisi et al. [27] | | | | | | | | | | | |
| Tiittala et al. [28] | | | | | | | | | | | |
| Williams et al. [29] | | | | | | | | | | | |
| Monge-Maillo [48] | | | | | | | | | | | |
| Salas-Coronas et al. [53] | | | | | | | | | | | |
| Laganá et al. [38] | | | | | | | | | | | |
| Kissinger et al. [49] | | | | | | | | | | | |
| Barnett et al. [39] | | | | | | | | | | | |
| Kumar et al. [30] | | | | | | | | | | | |
| Stauffer et al. [31] | | | | | | | | | | | |
| McCarthy et al. [32] | | | | | | | | | | | |
| Ditton & Lehane [40] | | | | | | | | | | | |
| Wiwanitkit & Waenlor [41] | | | | | | | | | | | |
| Martinez-Donate et al. [50] | | | | | | | | | | | |
| Manzardo et al. [44] | | | | | | | | | | | |
| Jackson et al. [51] | | | | | | | | | | | |
| Crawshaw et al. [33] | | | | | | | | | | | |
| Seagle et al. [52] | | | | | | | | | | | |
| Sudhinaraset et al. [42] | | | | | | | | | | | |
| Tramuto et al. [43] | | | | | | | | | | | |
| Coppola et al. [34] | | | | | | | | | | | |
| High risk of bias, n (%) | 16 (33) | 16 (33) | 10 (20) | 9 (18) | 7 (14) | 8 (16) | 10 (20) | 0 (0) | 26 (53) | 20 (41) | 1 (0,2) |

■ = low risk of bias, SUM 0–3 points.　　☐ = intermediate risk of bias, SUM 3–6 points.　　■ = high risk of bias, SUM 7–10 points.

## 4. Discussion

This systematic review found that the highest estimated HIV prevalence existed among migrants originating from Southern Africa and the lowest in migrants originating from the Middle East. The Migration/Origin ratio varied significantly depending on country of screening. The overall Migration/Origin ratio was >1, indicating that the average HIV prevalence among migrants was higher than that of the general population in the country of origin. Further, HIV prevalence among migrants originating from high endemic countries was generally higher than that of the autochthonous population.

### 4.1. Practical Implications

The findings of this study have important clinical implications, as the prevalence of internationally displaced people has been rising steadily during the last decade, and migrants constitute an increasing population seen in healthcare facilities [2]. A notable difference in prevalence among migrants and the autochthonous population in the country of origin was seen, for which there may be several reasons. While migration itself is not a risk factor for HIV, specific factors associated with migration increase the risk of HIV. Many migrants spend long periods in transit, often in several countries with different disease epidemiology, sometimes in crowded refugee camps where communicable diseases are easily spread, making them vulnerable to infectious diseases [10]. Further, reasons for migration and screening initiatives could influence the difference in HIV prevalence among migrants and autochthonous populations. The possibility that significant numbers of HIV-positive patients migrate to foreign countries to receive optimal treatment, resulting in an increased prevalence of HIV among screened migrants, should be considered [66].

Besides, it is possible that screenings of migrants for HIV in receiving or transit countries are more comprehensive than in many countries of origin, resulting in more accurate prevalence estimates among migrants than among autochthonous populations.

It is likely that socioeconomic status is associated with the risk of HIV infection. In the migration context this is, among others, expressed as migrant status and health knowledge. Migrants in unfavorable economic positions and with little knowledge on health and risks of being infected with sexually transmitted diseases are likely to be particularly vulnerable to HIV infection [67–69]. Social and cultural disparities towards HIV means that being diagnosed with HIV is often connected to shame and stigma and accompanied with a complex life situation in many populations, from where migrants originate [70]. Studies have found that lower adherence and retention to HIV care among migrant groups is an emerging problem. This complicates both diagnosing and treatment, resulting in some migrants living in secrecy with their HIV diagnosis, hence not receiving crucial treatment, which often leads to a reduced quality of life both physically and mentally [71].

The findings of this study indicate that migrants originating from high endemic countries, especially countries in East-, West- and Southern Africa, respectively, could to a great extend benefit from systematic screenings for HIV. Contrary, HIV screenings of migrants originating from low endemic countries, especially countries in Eastern Europe and the Middle East, might not need to be of as high priority due to generally low prevalence estimates. However, systematic screening for HIV when arriving in a recipient country is important as prevalence is often higher among migrants than among autochthonous populations. The systematic screenings should be conducted to diagnose and treat diseases, thus contributing to a better health and quality of life of migrants [72,73]. In addition, more international focus should be put on migrant health, HIV risk factors and discrimination, as well as the importance of and possibilities for efficient HIV treatment options for migrants in receiving countries. This should be done to prevent the increased HIV prevalence in several migrant groups and to improve health and quality of life for migrants in general.

### 4.2. Limitations of This Study

When comparing the yields of the included articles in this study to the WHO estimated HIV prevalence, some reservations must be taken into consideration. Firstly, it must be considered that the data provided by WHO solely estimates prevalence among adults aged 15–49 years, whereas data included in this study encompass HIV prevalence in both adults and children. Secondly, the validity of methods of measurement by WHO estimated prevalence varies. In some countries, the estimates are based on data generated by surveillance systems that focus on pregnant women who attend sentinel antenatal clinics. In other countries, estimates are based on nationally representative serosurveys. Hence, an erratic quality in the methods of measurement must be taken into account [74]. Thirdly, data included in this study is obtained in the time span from 1993 to 2020, thus creating some uncertainty when compared to WHO estimates from 2018–2019.

The awareness of bias in the included studies was crucial to evaluate the validity of the estimated results. The varied time span from migrants arriving at the recipient countries until being screened for HIV was a limitation to this study. Migrants who have spent a long time in a recipient country before being screened for HIV have an increased risk of having acquired this communicable disease post migration. However, due to variable amounts of time spent in recipient countries before being screened for HIV in the included studies, an accurate assessment of the percentage of migrants who possibly acquired HIV post migration cannot be made [75,76]. A great heterogeneity of available data on HIV prevalence based on country of origin and the sizes of the study populations also increase the risk of bias and creates a limitation to this study. In some of the included countries of interest, very limited amounts of data on migrants exist. This results in small study populations and consequently inappropriate denominators for the parameter of interest, causing an increased risk of bias. Examples hereof are Guinea-Bissau, Malawi, Nicaragua,

and Poland. Only articles published in English were included in the study, which possibly leaves out published literature on the topic written in other languages.

## 5. Conclusions

The yields of this study suggest that HIV prevalence among many migrant groups differs from that of the autochthonous population, and that the prevalence among migrants originating from countries with a high HIV prevalence generally is higher than that of the autochthonous population. The highest estimated HIV prevalence existed among migrants originating from Southern Africa and the lowest in migrants originating from the Middle East. In a clinical setting, this should be taken into consideration by health authorities in recipient countries where systematic screenings are not an option, when allocating migrant groups to specific health screenings. However, more data on the topic is necessary to make sufficient estimates covering migrant groups originating from all parts of the world.

**Author Contributions:** Conceptualization, C.S. and C.W.; methodology, C.S. and C.W.; software, C.S.; validation, C.S. and C.W.; formal analysis, C.S.; investigation, C.S.; resources, C.W.; data curation, C.S.; writing—original draft preparation, C.S.; writing—review and editing, C.W.; visualization, C.S. and C.W.; supervision, C.W.; project administration, C.S.; funding acquisition, not relevant. All authors have read and agreed to the published version of the manuscript.

**Funding:** This research received no external funding.

**Institutional Review Board Statement:** Ethical review and approval were waived for this study, as data was extracted from already published literature in which all included humans were anonymous and data was obtained legally and with consent.

**Informed Consent Statement:** Patient consent was waived in this systematic review as all included data has been obtained with consent and all included patients are anonymous.

**Data Availability Statement:** Data available in a publicly accessible repository.

**Conflicts of Interest:** The authors declare no conflict of interest.

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
