# Peer review of "HIV Prevalence in Migrant Groups Based on Country of Origin: A Systematic Review on Data Obtained between 1993 and 2020"

_sustainability, doi:10.3390/su132111642_

Round 1

Reviewer 1 Report

This is an important and current subject regarding HIV and migration. The authors include literature up to 2020 and was expecting a comment made on COVID-19/

MAJOR COMMENTS

  1. The authors mention internally displaced people as if similar to cross boarder migrants (international migrants). Please define internally displaced and be clear on the focus of the paper. This report provides a clearer scope on migration: "International Organisation for Migration (IOM). World Migration Report 2020. Geneva, Switzerland: IOM; 2020."
  2. The authors state two aims in the abstract: a) to investigate the HIV prevalence in migrant groups based on country of origin and b) to compare this to the WHO estimated prevalence for the countries of interest. Please clarify the "countries of interest". I am assuming these are countries of origin. Additionally, the results presented are based on region of origin as opposed to country of origin. May you please be clear.
  3. The abstract states that 49 studies were included after screening by tittle and/or abstract. Does it mean that full manuscripts were not screened? This is not consistent with the main text.
  4. The abstract does not define migration/origin ratio.
  5. What years were covered in the literature search. Please revise and be consistent. What was the purpose of: "In the second search the year of publication was adjusted so that only articles published in 2020-2021 were screened."

MINOR

Abstract: The abstract appears to have different fonts used. Please revise.

Please add the years covered by the literature search in the abstract. It is confusing in the main text.

Paragraphs need to be revised and be consistent of indenting.

References need to be revised. Some references do not have clear dates of when cited e.g. 3rd reference

Reviewer 2 Report

This study by Schousboe C and Wejse C. investigate the HIV prevalence in migrant groups based on country of origin and comparing to the WHO estimated prevalence. They have conducted an extensive literature search in Pubmed and Embase using certain keyword “HIV infection”, “migrants”, refugees. Based on the search, 2345 records have been identified and then using exclusion and inclusion criteria, 49 literature have been used to conduct this systematic review.

The authors have performed this analysis and presented well. Few areas need to be taken care of

  1. In the title, please mention the year range like “from the data obtained in between1993-2020” (in the selection criteria, the authors have used/ selected the publication in between 1993-2020).
  2. Explain the method section “2.3 risk of bias assessment” more in details.
  3. Please provide line no 349-358 as a dedicated section “limitation of this study”.
  4. Kindly revisit the reference formatting.

Round 2

Reviewer 1 Report

No further queries or comments.

Author Response

Dear Editor,
Thank you for the relevant comments.
We have now created a new section with the sub-heading "Practical Implications" to the discussion (lines 371-413, 525 words) where the findings of the study are being reflected on and put into context of migrant health and quality of life.

Kind regards,
Cecilie Schoubsoe,
Corresponding author
